# Acupuncture for Pain Management in Pediatric Patients with Sickle Cell Disease

**DOI:** 10.3390/children9071076

**Published:** 2022-07-19

**Authors:** Krystal S. Plonski

**Affiliations:** Department of Anesthesiology and Pain Medicine, Seattle Children’s Hospital, Seattle, WA 98105, USA; krystal.plonski@seattlechildrens.org

**Keywords:** acupuncture, pediatric pain, acute vaso-occlusive episode, integrative pain medicine, pediatric pain management, sickle cell disease, sickle cell pain episode, vaso-occlusive pain

## Abstract

Pain management in an acute vaso-occlusive episode for pediatric patients with sickle cell disease (SCD) is challenging and often is focused on opioids, IV fluids, regional anesthesia, ketamine infusions, and non-steroidal anti-inflammatory drugs (NSAIDs). Acupuncture has long been studied as an effective method of pain relief, although the use of acupuncture in pediatric patients with SCD during an acute vaso-occlusive pain episode is vastly understudied. This article provides a review of current research regarding the use of acupuncture as a pain treatment strategy for pediatric patients with SCD experiencing acute pain. A literature review of scientific papers published within the last ten years was conducted on the topic. Five primary literature articles on acupuncture for pain management in pediatric patients with SCD were reviewed. Acupuncture is feasible and acceptable, with statistically significant findings for effectiveness as an adjunct treatment for pain in this setting. It is concluded that acupuncture is a promising and understudied therapy for the treatment of pain during an acute pain episode in pediatric patients with SCD. Hopefully, this paper stimulates interest in this specific area of medicine and prompts future research studies to be conducted to reveal conclusive outcomes.

## 1. Introduction

Pain management during an acute vaso-occlusive episode in a pediatric patient with sickle cell disease (SCD) is challenging to treat and standard-of-care options are limited to opioids, IV fluids, adjunct ketamine infusions, regional anesthesia, and nonsteroidal anti-inflammatory drugs (NSAIDs) [1,2,3,4]. Acupuncture has long been studied for pain relief, however, a unified theory of its action in Western Medicine has not been accepted [5,6,7,8,9,10,11,12,13]. Moreover, acupuncture for the use of pain management in pediatric patients with SCD during an acute pain episode is a vastly understudied area of medicine [5,6,7,8,9,10,11,12,13].

The purpose of this review article is to examine the existing literature regarding the use of acupuncture as an adjunctive pain treatment option in pediatric patients with SCD pain episodes. Much research potential remains for adjunct integrative pain treatment options in the pediatric population with SCD for both acute and chronic pain [5,6,7,8,9,10,11,12,13].

### 1.1. Sickle Cell Disease

Sickle cell disease is an inherited hemoglobinopathy causing an alteration in red blood cell shape, motility, and function [1,2,3,4]. Scientists have defined four major pathological processes of SCD: hemoglobin S polymerization, vaso-occlusion, hemolysis mediated endothelial dysfunction, and sterile inflammation [2]. The result is a disease process that creates anemia, systemic inflammation, oxidative stress, tissue and end-organ ischemia, bone and/or bone marrow destruction, and pain [2,3,4].

Estimates, for both adult and pediatric data, show that 176,000 people die of SCD-related complications per year in the United States [2]. Despite mortality rates vastly improving over the years, patients with SCD have a life expectancy 2–3 decades shorter than the general population [3]. Comorbidities for pediatric patients with SCD include risks of infection, acute chest syndrome, stroke, splenic sequestration, acute anemia, nephropathy, acute renal failure, retinal damage, cholecystitis, priapism, and avascular necrosis [3]. To date, disease-modifying treatments such as hydroxyurea or blood transfusions, target primary and secondary stroke prevention, which reduce the frequency of sickle-cell related pain and acute chest syndrome [3]. Researchers and clinicians alike hope that hematopoietic stem cell transplantation will gain more traction and accessibility [3].

From infancy, the most common symptom of SCD is pain, which can present as acute intermittent pain, chronic daily pain, and acute on chronic pain [4]. The standard-of-care pharmaceutical pain management strategy in children is dependent on the type of pain experienced, but also focuses on IV fluids, immediate or sustained release opioids, NSAIDs, adjunct ketamine infusions, and regional anesthesia [4]. These treatments do not come without significant side effects for patients, including constipation, pruritis, urinary retention, and nausea; many of these may require additional medication management or promote the long-term potential to develop altered pain processing, such as central sensitization [4,14]. Second-line pharmaceutical treatments, such as serotonin and norepinephrine reuptake inhibitors (SNRIs), tricyclic antidepressants (TCAs), and gabapentinoids, are also used when additional components to a pain experience need to be addressed [4]. Certain integrative treatments, such as cognitive behavioral therapy, are supported by pediatric pain management guidelines, but pediatric massage and acupuncture lack the rigorous study design to create conclusive recommendations for these therapies [4].

### 1.2. Acupuncture

Acupuncture is a worldwide system of medicine that dates to 100 B.C. in China from the medical writings in the *Huang di Neijing* (The Yellow’ Emperor’s Classic of Medicine) [15,16]. Acupuncture uses sterile, thin, solid, and disposable stainless-steel needles inserted into specific points on the body to obtain a therapeutic effect [17,18,19,20,21,22]. Eastern Medicine philosophy views acupuncture as the movement of energy (*qi*) and blood through specified body points and pathways; these are expressed in the core theories of Yin/Yang, Five Elements, and Zang Fu Organ Theory [17,18,19,20,21,22]. Acupuncture uses observational information from the appearance of the patient’s tongue and the feeling of their pulse, in addition to patient questioning, to develop a diagnosis and treatment strategy [17,18,19,20,21,22]. Additionally, Traditional Chinese Medicine (TCM) encompasses other therapeutics besides acupuncture, such as acupressure, *tuina* (massage), *guasha* (cutaneous scraping technique), and herbal formulations [17].

Acupuncture is generally a safe and effective treatment in both adult and pediatric patients when administered by a board-certified acupuncturist [22]. In 2011, The American Academy of Pediatrics (AAP) conducted a large systematic review of safety events during pediatric acupuncture visits between June 2007 and September 2010 [23]. They found that mild and non-worrisome adverse events during or after an acupuncture session included bruising, bleeding, mild pain upon needle insertion, and fatigue [14,23,24]. More significant and worrisome effects were rare and far more likely when acupuncture was administered by poorly trained practitioners, lacked clean needle techniques, or the treatments did not follow the standard of practice [14,23].

To date, Western Medicine has not been able to identify a single unifying mechanism of action for acupuncture for pain control. Research indicates acupuncture may have local and distal actions, as well as influences on human biochemistry, the nervous system, circulatory system, immune system, and fascial networks [17,18,19,20,21,22,25,26,27,28,29,30,31].

Based on SCD pathophysiology, four mechanisms of action may be theorized for how acupuncture impacts SCD pain: vasodilation, modulation of endothelial dysfunction, anti-inflammatory effects, and the release of endogenous opioids [2,9,21,29,30,31]. Unfortunately, no current studies exist indicating reduction in hemoglobin S polymerization through acupuncture. Research has shown that acupuncture can increase local concentrations of nitrate plus nitrite end products and reliable indicators of nitrous oxide metabolism and formation at acupuncture points resulting in local vasodilation and improvements in blood flow [21]. Furthermore, acupuncture has also been shown to improve endothelial dysfunction in other clinical conditions, such as hypertension, although not specifically demonstrated in SCD [9,21,29]. A recent paper reviewed decades of research, both in human and animal models, dedicated to acupuncture’s anti-inflammatory effects through peripheral nerve modulation as a potential treatment for organ dysfunction in sepsis [31]. Both acupuncture and electroacupuncture have effects on downregulating pro-inflammatory cellular signals, such as tumor necrosis factor-alpha, interleukin-1, and interleukin-6 in mice and human models [31]. Lastly, acupuncture’s analgesic mechanism of action is initially through point stimulation in a muscle (i.e., stimulating afferent A-delta and C fiber signaling in muscles), resulting in a local release of endogenous opioids (e.g., dynorphin and enkephalins) in the spinal cord to the midbrain [30]. Midbrain synapses later trigger efferent signals propagated by serotonin, dopamine, and norepinephrine back onto the spinal cord to inhibit and suppress pain transmission signals [30]. While intriguing and plausible, these notions remain heuristic at present, as none of these theories have been specifically validated in patients with sickle cell disease.

Current studies on the clinical efficacy or effectiveness of acupuncture for SCD-related pain in children consist of retrospective chart reviews or feasibility studies on the acceptance of acupuncture. They lack the methodological rigor needed to assess acupuncture’s benefits [5,6,7,8,9,10,11,22,32]. In 2020, the American Society of Hematology (ASH) set out guidelines for the management of both acute and chronic pain in the adult and pediatric patient population, concluding a lack of evidence on the recommendation for or against the use of acupuncture for the treatment of acute and/or chronic pain in the pediatric patients with SCD [4]. Additionally, the American Academy of Pediatrics (AAP) put forth a statement indicating that opioids are commonly used in pediatric patients with SCD for pain management [22]. Acupuncture is recognized by the NIH as generally safe and effective for pain, and acupuncture appears to be a promising therapy for pediatric patients with SCD both in inpatient and outpatient settings [22]. To echo the ASH recommendations, the AAP statement lacks formal recommendations for acupuncture in pediatric SCD pain episodes as a useful co-treatment option due to a paucity of high-quality evidence-based research [22].

## 2. Materials and Methods

A literature review of scientific papers published between 2011 and 2021 was conducted on the topic of acupuncture or acupressure as a treatment for pain in pediatric patients with SCD in the Ovid Medline, EMBASE, and CINAHL databases. A search on concepts specific to acupuncture, acupressure, and sickle cell anemia was completed using the main keywords: “acupuncture”, “acupuncture therapy”, “acupressure”, “sickle cell anemia”, “anemia”, “sickle cell”, “acupuncture analgesia”, “pain”, and “pain management”. Exclusion criteria included: articles published prior to 2012, children >21 years old, languages other than English, and specified publication types (e.g., comments, conference materials, dissertations, editorials, and letters). For more detailed search criteria, please see Expanded Search Strategies in Appendix B.

Nine results were obtained, and the removal of four additional non-relevant citations occurred. Five papers remained from the original search indicating primary research for the use of acupuncture for acute pain management in the pediatric sickle cell disease population in this paper (Table 1).

## 3. Results

The first article is from 2015 where Tsai et al. published a case report of a patient with SCD (genotype undisclosed) who received acupuncture treatment 58 times in the inpatient, outpatient, and pediatric Emergency Department settings over the course of three years [5]. All treatments were done by either a licensed acupuncturist or a pediatric emergency room medical physician/acupuncturist following TCM Acupuncture theory [5]. Acupuncture treatments were variable in the inpatient setting based on patient availability and interest but were intended to be more structured in the outpatient setting with a goal of once weekly treatments ×7 weeks [5]. Patient compliance with follow-up for maintenance acupuncture treatments was a focus in the outpatient setting [5]. All acupuncture treatments were performed with sterile and disposable needles (averaging 14.5 needles per treatment), and needle retention ranged from 20–30 min per treatment [5]. Examples of acupuncture points selected in all studies are detailed in Appendix A [5,6,8,9]. No adverse events attributable to acupuncture treatments were reported during this single patient retrospective case review [5].

The authors highlight that acupuncture was reportedly useful for this patient when experiencing severe pain and often also when medications were not helpful [5]. Two types of pain scales were used, a 0–10 (11 point) numerical rating scale (NRS) or a 0–4 (5 point) Wong–Baker FACES scale [5]. Statistical significance of pain improvement in the NRS for children with SCD is reported as a decrease of 0.9 points [22]. The authors report this patient’s mean decrease in pain score as 2.1 across all treatment settings [5]. Interestingly, the acupuncture providers documented occurrences of the patient falling asleep after needle insertion, which the authors attributed to an anxiolytic effect and is an infrequently acknowledged benefit of acupuncture treatments for pediatric patients in sickle cell pain episodes [5]. Although this study indicated some feasibility and acceptability of acupuncture in both the inpatient and outpatient setting, the limitations of this study included a single patient perspective, retrospective design, bias of treatment selected, use of inconsistent pain scales, extrapolation of statistical significance from a single case, non-uniform timing of treatments, and an occasional lack of pre/post-treatment pain assessments, which could be remedied in any future prospective study design [5].

In the outpatient setting, TCM acupuncture point selection was focused on wellness and current symptoms, while inpatient treatments were focused on pain [6]. Acupuncture needles were used for 15–20 min per session without additional TCM therapies employed [6]. Based on provider preference, additional press transdermal needles may have been used at any acupuncture point and retained for 24–48 h post-acupuncture treatment with removal by the patient, caregiver, or nurse upon their follow-up appointment [6]. One adverse event was reported as one participant’s skin was scratched by an acupuncture needle, with a request to discontinue treatments [6]. However, otherwise, all treatments were tolerated well without additional adverse events [6].

The researchers reported descriptive statistics for these patients as having an average age of 16.85 years, with 75% identifying as female and the most common genotype being HbSS (83%) [6]. Home medications included hydroxyurea in 58% of participants, 66% reported maintenance pain medications (gabapentin, methadone, or long-acting opioids), and 100% of participants were prescribed short-acting opioids to be taken as needed [6]. Ninety-two percent of all participants tolerated acupuncture well, and 58% received more than one treatment [6]. Of the 11 remaining patients at the end of the study, 100% were found to report a good experience with acupuncture, and 73% specifically expressed improvement in pain post-treatment [6]. Of the sessions that included pre/post-treatment pain scores, the researchers found that an average difference of 0.933 points with a 1.03 standard deviation of pain reduction was statistically significant, meeting Myrvik’s threshold on the NRS (*p* < 0.000) [6,22]. The strength of this study indicates the feasibility of acupuncture in the inpatient and outpatient setting as well as potential interest for ongoing treatments despite a retrospective design, lack of consistent pain scoring pre/post-treatment, absence of opioid use data, and a lack of a control arm [6].

The third study, also by Mahmood et al., included a prospective single-center pilot study of 31 patients ten years and older for the management of chronic pain [7]. Of the integrative medicine options, acupuncture was included among pain pharmacology services, psychology, massage, aromatherapy, and healing touch [7]. The Treatment Evaluation Inventory-Short Form (TEI-SF) was administered, which assessed the overall experience after all integrative medicine modalities were offered. As a result, 72% of patients agreed that the therapies were acceptable for treating adolescent pain, 84% of patients indicated they were willing to use the procedures to deal with pain, 88% had a positive experience, and 80% agreed that these modalities should only be used with an adolescent’s consent [7]. Regarding adverse events, no specific acupuncture events were reported. However, based on TEI-SF results, 32% of participants indicated that a patient could experience discomfort during any treatment [7]. This study had many limitations, with the primary problem being that each individual integrative modality was not teased apart from the rest. No conclusions can be drawn regarding acupuncture (or another integrative treatment) on its effect on pain responses. Lastly, positive responses from patients and parents may be due to an underlying preference for these therapies. From this study, no significant conclusions regarding acupuncture alone can be deduced. However, the indication of high patient and parent interest in various integrative medicine modality options for children and adolescents in pain is promising [7].

The fourth study, by Reece-Stretman et al. was a prospective feasibility study to determine if acupuncture would be accepted as an adjunct treatment to standard-of-care pain management pediatric patients with SCD who were hospitalized for management of acute pain [8]. The researchers’ secondary aim was targeted at whether acupuncture could help to reduce pain scores in these patients and/or affect length of hospital stay [8].

Within 24 h of inpatient admission, pediatric patients eight years and older with any SCD genotype were approached-alongside parental consent-to determine interest in receiving acupuncture in addition to standard-of-care pain management (IV fluids, opioids, anti-inflammatory agents) [8]. Acupuncture treatments were offered by three acupuncturists (one licensed acupuncturist and two pediatric anesthesiologist/acupuncturists) within the first 24 h of admission following TCM diagnosis and point selection [8]. Sterile, long, disposable, stainless steel acupuncture needles of various sizes were used, without electrical stimulation, with an average of 6–8 needles per treatment and retained for 10–15 min. Pyonex press tack needles were also used individually for patients based on provider choice for a few hours to one day after treatment [8]. Once daily sessions of acupuncture for the duration of their inpatient hospital stay were offered, and if an acupuncturist was unavailable, their information served as a control patient for the day [8].

Information regarding clinical and laboratory data, length of stay, opioid use, and readmission within one month was collected. Temporal associations with opioid use related to acupuncture were not able to be extracted, but total opioid use was extracted via chart review [8]. Patients and parents completed the TEI-SF survey to evaluate acceptability of acupuncture as a treatment option [8]. The acceptability rate of acupuncture was 66%, with a 95% confidence interval, which closely compares to their hypothesis of 60% acceptability [8]. All patients had as-needed short-acting opioids at home, and all, but one, were prescribed oxycodone; in this case, hydromorphone was prescribed instead [8]. The researchers indicate a statistically significant pre-/post-pain score reduction in the treatment group compared to the non-treatment group, with a difference of 1.33, or a 19.4% reduction (*p* < 0.0001) [8]. No significant adverse events were reported for the treatment or non-treatment group in this study [8].

A nonsignificant trend in total length of hospital stay was shorter (median of 3.74 vs. 4.6 days), and readmission within 30 days was lower (15% vs. 30%) in the treatment group compared to controls [8]. Milligram equivalents (MME/kg/day) were also higher in the treatment group (1.31 vs. 0.99) but not at statistical significance [8]. When asked to answer questions on the TEI-SF regarding acupuncture, 95% (18/19 patients/parents) agreed to the survey; 89% of participants agreed that acupuncture was an acceptable way of treating pain, 94% were willing to use it to manage adolescent pain, 67% agreed that it should only be offered with an adolescent’s consent, and 100% of patients reported a positive experience with acupuncture [8]. The strength of this study exemplifies the feasibility and acceptability of acupuncture in addition to standard-of-care pain management in the inpatient hospital setting. However, limitations include the sample size and statistical significance relating to clinical significance in pain score reduction for patients [8]. Furthermore, questions remain regarding the increased trend for MME in the acupuncture treatment group compared to control [8].

Lastly, the fifth study is by Tsai et al., where a retrospective chart review was conducted on 24 patients with SCD receiving standard pharmacologic pain management for a total of 90 acupuncture treatments across all participants [9]. Acupuncture was administered by six licensed acupuncturists using both TCM and Japanese-style acupuncture without electrical stimulation [9]. Sterile, single-use, disposable SIRIN acupuncture needles of various lengths, gauges, and needling depths were used based on practitioner preference and acupuncture style [9]. Adverse events were defined as pain, bleeding or bruising at the needle site, dizziness, or syncope [9]. No adverse events were recorded [9].

The researchers conducted 48 acupuncture treatments in the outpatient setting and 42 acupuncture treatments in the inpatient setting, with an average length of treatments of 18.5 min and the average number of needles used per patient as 6.8 [9]. The median age for participants was 17.5 years. Additionally, 62% were female, 37.5% were African American, 50% were Hispanic, and 12.5% were of another ethnicity [9].

Fifty-five specific acupuncture treatments directed for pain mostly located in the back, non-specific “whole body”, legs, abdomen, arm, chest, head, jaw, and shoulder also included pre/post-pain scores documented based on either the 0–4 pain scale (42 treatments) or the 0–10 pain scale (13 treatments) [9]. The researchers report that acupuncture treatments effectively decreased mean pain scores by one point, regardless of the pain scale used, with more statistically significant results using the 0–4 pain scale, possibly due to sample size [9]. Additionally, of the 55 treatments with pain scores, almost 66% of patients also reported both relief in pain and anxiety after an acupuncture treatment, with documentation of “feeling better” or “much calmer with less pain” [9]. A significant limitation of this study is that the acupuncture practitioners possibly used other non-pharmacologic therapies they hold certifications in, such as aromatherapy, massage, and Reiki, during a session [9]. The researchers did not delineate which treatments were and were not used in addition to acupuncture. Therefore, due to the possible concurrent therapeutic options during a session, it is difficult to draw conclusive remarks regarding the use of acupuncture alone for pain management [9].

## 4. Discussion

Pain is a common, subjective, and complex experience for the pediatric patients with SCD and can be a challenging symptom to treat for a provider, particularly when opioids and other medications are ineffective. Acupuncture has been studied for its multifactorial physiology of pain management and the NIH and AAP support the potential use of acupuncture for certain pain conditions, both in adults and in children. In addition to the standard-of-care management of acute sickle cell pain episodes in the inpatient and outpatient setting, a multidisciplinary approach that includes acupuncture can provide potential treatment options for difficult-to-manage pain. Acupuncture, when applied by a well-trained and board-certified acupuncturist, remains a safe therapeutic with few adverse effects. Currently, acupuncture remains an individualized and potential treatment to help reduce pain in pediatric patients with SCD.

This review article aimed to examine the existing literature regarding the use of acupuncture as a pain management option, in addition to pharmacologic standard-of-care, in pediatric patients with SCD pain episodes. Of the five papers examined, all acupuncture treatments in study participants were in addition to standard-of-care pain management rather than in place of it. Literature search results also showed no specified set of SCD-specific or pain-specific acupuncture points for patients. Instead, the authors highlighted the most used points during an acupuncture treatment. Appendix A outlines which points are specifically mentioned for each research article. Zero results populated for research articles on acupressure or electroacupuncture for pediatric patients with SCD-related pain. As detailed in the text, all acupuncture treatments lasted between 10–30 min. No significant adverse events occurred in any study participant that received acupuncture treatments in the studies examined, except for a report of a mild adverse event of a superficial skin scratch in one study.

The primary barrier for applying acupuncture therapy to routine pain treatment in pediatric patients with SCD remains a lack of rigorous research design, making it difficult to determine clinical best practice standards. Inherently, acupuncture is difficult to study due to the nature of controlling for needle insertion versus no needle insertion, sham acupuncture (common placebo method), and consent for randomization of the treatment in with pediatric patient populations. Of the studies available, many limitations exist within study design. However, the positive results of these studies indicate that acupuncture is both accepted, feasible, beneficial, and without significant side effects in the pediatric emergency department, inpatient, and outpatient clinic settings. Additionally, it is clear from the results of this literature review that in addition to standard-of-care treatments, families are interested and open to non-pharmaceutical pain management for their children with SCD.

In clinical settings, limitations of inclusion of acupuncture therapy for pediatric patients with SCD include lack of highly trained pediatric acupuncturists, lack of specialty training for SCD among acupuncturists, provider bias of pediatric patients having ‘needle phobia,’ pediatric patient’s actual ‘needle phobia,’ and lack of knowledge for referring providers for benefits of acupuncture in SCD pain. In hospital settings, limitations of implicating acupuncture therapy include staff acupuncturists, the timing of treatments, patient interest, patient availability, and funding for acupuncture programs. More specific SCD-related limitations and disease process comorbidities may also pose inherent barriers to care, which include physical limitations from pain, feelings of fatigue, feelings of stigma, and emotional responses to health care visits for pain management, interpretation of acupuncture as an invaluable, and interpretation of acupuncture as an experimental treatment may provide additional barriers [13]. Lastly, patient logistical barriers include limitations with insurance coverage, insurance reimbursement, transportation, and access to highly trained pediatric acupuncturists in both urban and rural areas [13].

For the development of future studies, a prospective, randomized control trial with multicenter data and high patient enrollment following the Ped-IMMPACT Pain Guidelines and Standards for Reporting Interventions in Clinical Trials of Acupuncture (STRICTA) guidelines for acupuncture research would be ideal. Important areas of acupuncture research lacking for SCD include acupuncture’s effect on fetal hemoglobin levels, acupuncture’s effect on polymerization of hemoglobin S, SCD endothelial dysfunction, and substrates of nitrous oxide metabolism in SCD. Research opportunities for pediatric patients with SCD include: acupuncture as a maintenance or preventative strategy for the development of an acute sickle cell pain episode, validated surveys examining acupuncture as improving functionality during pain episodes and/or after pain episodes, changes in pain tolerance, body map diagrams of pain experiences, consistent pain scoring pre/post-treatment, total opioid use comparing treatment to non-treatment groups, time of opioid use to when acupuncture is administered, length of time acupuncture treatments keep effects, close monitoring of adverse pre-/post-treatments, and total length of hospital stay. More difficult areas of acupuncture research relate to patient/parent interviews on knowledge of acupuncture for pain in SCD, the cost-effectiveness of acupuncture treatments, patient and/or parent treatment satisfaction, and improvement in quality of life for both patients and/or caretakers. Many opportunities regarding the use of acupuncture as a treatment option for pediatric sickle cell pain episodes remain unexplored, and results could potentially and significantly influence clinical best practice guidelines for not only these patients but also their families.

## 5. Conclusions

After reviewing the current literature on the topic, acupuncture remains a promising and understudied therapy for the treatment of acute pain during an acute vaso-occlusive pain episode in pediatric patients with SCD. This author hopes that this paper stimulates interest and justification in this specific area of study to encourage future additional prospective research studies to help us answer this question and advance non-pharmaceutical treatment options for this patient population.

## Figures and Tables

**Table 1 children-09-01076-t001:** Summary of Research Articles on the use of Acupuncture for Pain Management in Pediatric Patients with Sickle cell Disease.

	Title & Author/Year	Type of Study, Location, and Duration	Number Participants, Demographics, Phenotype, Inclusion/Exclusion Criteria	Diagnosis and Treatment Used	Points Selected, Equipment & Duration of Treatment	Outcomes Measured & Results
*1*	Acupuncture for Sickle Cell Pain Management in Pediatric Emergency Department, Hematology Clinic, and Inpatient Unit By Tsai et al., 2015 [5]	Retrospective chart review from 1 patient with SCD in the outpatient, inpatient, and ED setting	1, N/A	TCM tongue and pulse diagnosis, Acupuncture traditional needles	TCM Acupuncture point selection on intent to treat pain/anxiety + underlying constitutional imbalance, pre-sterilized, disposable #3 SEIRIN needles, 0.20 × 30 mm for body points and 0.20 × 15 mm for auricular points following clean needle technique (CNT); Needling depth: 0.5–1 cun (approx. ½ inch); Needle retention 20–30 min; Range of acupuncture needles 3–27 with an average of 14.5 per tx; Example of acupuncture points used: LI11, LI4, ST44, LV3, ST36, SP6, KD3.	Total of 58 treatments, ×3 years; Self-reported patient or parental reduction in anxiety or patient falling asleep; Pain scores collected via charted NRS or Wong-Baker FACES with a mean decrease in pain across all treatment settings: 2.1 points. No adverse events reported
*2*	Acupuncture for pain management in children with sickle cell disease By Mahmood, et al., 2020 [6]	Single academic pediatric center experience; inpatient/Outpatient facilities; IRB approved	12, average age of participants 16.85 years, 75% female; most common genotype HbSS (83%), median number of hospitalizations in the previous year was 8 (1–19)	TCM diagnosis, Acupuncture with traditional and press tack needles retained 24–48 h	TCM Acupuncture point selection in outpatient setting focused on wellness + current symptoms and in the inpatient setting focused on pain; Use disposable one time use stainless steel needles 0.12–0.16 mm diameter were used; Needle retention 15–20 min; Number of needles ranged from 2–10 (most used: LV3, LI4, DU24.5, DU20); Pyonex press tack needles (stick-on intradermal needles) were applied to some acupuncture points based on provider preference	33 total acupuncture treatments × 2 years Pain scores, patient level descriptors collected. Where pre/post pain scores were collected (15/33 sessions), the mean pre score was 6.17 +/− 2.73 and the mean post acupuncture score was 5.23 +/− 2.46, resulting in an average difference of 0.933 with a SD of 1.03 (*p* < 0.000) 1 adverse event (scratched by a needle)
*3*	Integrative holistic approaches for children, adolescents, and young adults with sickle cell disease: a single center experience By Mahmood, et al., 2021 [7]	Single center experience; outpatient integrative medicine clinic; IRB approved	31 patients, mean age 15, 67.7% female; 61.29% HbSS, 32.26% HbSC, 2% HbSβ	Psychology Healing touch Aromatherapy Acupuncture Mindfulness	N/A	80% of participants completed the Treatment Evaluation Inventory- Short Form (TEI-SF), which found that 72% of patients agreed that integrative therapies were an effective way to treat SCD pain, 84% of patients were willing to use these treatments for adolescent pain; 32% believe discomfort can occur, 72% believed these modalities to be effective
*4*	Acupuncture as an adjunctive treatment for pain in hospitalized children with sickle cell disease By Reece-Stretman, et al., 2021 [8]	Single center prospective pilot feasibility study, IRB approved	29 patients, 8+ years old, SCD all genotypes; Exclusion: inability to provide consent/assent, SCD or other medical complications, pregnancy/lactation	19 participants received acupuncture; 10 agreed to serve as controls	TCM Acupuncture point selection to address acute pain, enhance blood flow, and restore energetic balance; inpatient setting; needles 0.12–0.2 mm × 15–30 mm, treatment duration 10–15 min, no electrostimulation; LV3, LI4 most used; Pyonex press tack needles were kept on for several hours, up to 1 day per practitioner discretion	Reduction of pain scores (0–10 NRS) by 1.33 points (19.4%) were statistically significant, nonstatistical significance of a lowered length of hospital stay, readmission rates within 30 days, and increased MME/kg/day in treatment participants TEI-SF (18/19 completed) with 89% indicating acupuncture as an acceptable way of treating pain; 94% indicated acceptable for treating adolescent pain; 100% participants indicate a positive experience with acupuncture
*5*	Acupuncture for pediatric sickle cell pain management: A promising non-opioid therapy By Tsai et al., 2020 [9]	Retrospective chart review of a single institution experience, IRB approved	24 patients with SCD, median age was 17.5 years, 62% were female, 37.5% were African American, 50% were Hispanic, and 12.5% were of other ethnicity/race	TCM and Japanese style acupuncture with sterile needles with and without guide tubes; 48 treatments outpatient, 42 treatments inpatient	Sterile, single-use, disposable SEIRIN acupuncture needles, with and without guide tubes for insertion technique, of various lengths, gauges, and needling depths were used based on practitioner preference; mean treatment duration 18.5 +/− 4.8 min (10–30 min) with mean 6.8 needles per patient (1–21 needle range)	Primary outcome: 4-point and 10-point verbal pain scores: Other information collected: demographics, pain location, adverse events

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
