# Peer review of "Acupuncture for Pain Management in Pediatric Patients with Sickle Cell Disease"

_children, 2022, doi:10.3390/children9071076_

Round 1

Reviewer 1 Report

I have no further comments.

Reviewer 2 Report

After revised, the manuscript explained clearly and easy reading. 

This manuscript is a resubmission of an earlier submission. The following is a list of the peer review reports and author responses from that submission.

Round 1

Reviewer 1 Report

The authors reviewed acupuncture as a promising and understudied therapy for the treatment of acute pain during an acute vaso-occlusive crisis (VOC) in pediatric patients with Sickle Cell Disease (SCD). This review paper stimulates interest in this specific area of medicine and encourages future research studies to be conducted to reveal conclusive outcomes for pediatric SCD.

Major Comments

  1. “Neuroimmunology plays a pivotal role in the development of an acute pain response in a SCD patient due to the interaction between mast cells, pro-inflammatory cytokines, neutrophils, and nerve cells in the dorsal root ganglion to activate pain pathways. Later, following an acute pain crisis, ischemia and reperfusion injury results in the persistence of activated peripheral nerve endings through these cellular and biochemical mediators. Furthermore, central sensitization can occur to form a predominant backbone of some chronic pain picture profiles in patients with SCD.” Therefore, I suggest that the author has better create a figure, which shows the possible mechanism of acute pain during an acute VOC in pediatric patients with SCD, and at the same time presents the possible action of acupuncture in a different part of the mechanism of this acute pain.

  1. Five papers were included in the final analysis of this manuscript. Please make a Table to show the features of each article.

Minor Comments

  1. Reference 2 and 14 are the same. Please check it.

Reviewer 2 Report

Submitted manuscript was felt very general, not found any new findings or scientific results.

Method and results was found critical errors, such as inclusion and exclusion criteria was not clear, therefore results was not understand well.

Reviewer 3 Report

In this manuscript, the author reviewed current research in the literature regarding the use of acupuncture as non-pharmaceutical pain co-management for pediatric patients with Sickle Cell Disease (SCD) experiencing an acute Vaso-Occlusive Crisis (VOC). The author concluded that acupuncture remains a promising and understudied therapy for the treatment of acute pain during an acute VOC in pediatric patients with SCD.

A few comments discussed in the following:

  1. Page 3, in Material and Methods section, please provide the key words used on the literature searching for this review.
  2. Page 3 to page 4 in Result section, please provide detail of the acupuncture treatments in each study, including the acupuncture points selected, the duration and frequency of the treatment, manual or electric stimulation etc.
  3. Page 4, please provide the information regarding the side effects and complication of acupuncture treatment from the above studies.
  4. Please provide a paragraph in discussion regarding the barriers to implicate acupunctures therapy to routine pain management in pediatric SCD patients.